# Amino acid residue at position 188 determines the UV-sensitive bistable property of vertebrate non-visual opsin Opn5

Chihiro Fujiyabu [1], Keita Sato [2], Yukimi Nishio[1], Yasushi Imamoto[1], Hideyo Ohuchi[2], Yoshinori Shichida[1,3] & Takahiro Yamashita [1✉]

Opsins are G protein-coupled receptors specialized for photoreception in animals. Opn5 is categorized in an independent opsin group and functions for various non-visual photoreceptions. Among vertebrate Opn5 subgroups (Opn5m, Opn5L1 and Opn5L2), Opn5m and Opn5L2 bind 11-*cis* retinal to form a UV-sensitive resting state, which is inter-convertible with the all-*trans* retinal bound active state by photoreception. Thus, these opsins are characterized as bistable opsins. To assess the molecular basis of the UV-sensitive bistable property, we introduced comprehensive mutations at Thr188, which is well conserved among these opsins. The mutations in Opn5m drastically hampered 11-*cis* retinal incorporation and the bistable photoreaction. Moreover, T188C mutant Opn5m exclusively bound all-*trans* retinal and thermally self-regenerated to the original form after photoreception, which is similar to the photocyclic property of Opn5L1 bearing Cys188. Therefore, the residue at position 188 underlies the UV-sensitive bistable property of Opn5m and contributes to the diversification of vertebrate Opn5 subgroups.

[1] Department of Biophysics, Graduate School of Science, Kyoto University, Kyoto 606-8502, Japan. [2] Department of Cytology and Histology, Okayama University Graduate School of Medicine, Dentistry and Pharmaceutical Sciences, Okayama 700-8558, Japan. [3] Research Organization for Science and Technology, Ritsumeikan University, Shiga 525-8577, Japan. ✉email: yamashita.takahiro.4z@kyoto-u.ac.jp

Animals utilize light as an important information source for various physiological functions. Opsins are universal photoreceptive proteins for both visual and non-visual photoreceptions in animals[1–3]. Opsins are characterized as G protein-coupled receptors specialized for photoreception and have seven transmembrane domains that bind the vitamin A-derivative retinal as a chromophore. Recent accumulation of the genomic information revealed that animals have various kinds of opsin genes, which are classified into several groups based on their sequences. Most opsins bind 11-*cis* retinal as an inverse agonist in the dark, and photoisomerization of 11-*cis* retinal to the agonist all-*trans* retinal leads to the formation of the active state of the opsin and its coupling with G protein.

Opn5 is the most recently identified opsin in the human and mouse genomes and forms an independent opsin group[4]. Opn5 genes have been identified in various vertebrates from fishes to primates and are classified into several subgroups[5,6]. Most mammals have only one Opn5 gene (Opn5m), whereas non-mammalian vertebrates have additional Opn5 genes, including Opn5L1 and Opn5L2. Previous studies revealed that the Opn5m and Opn5L2 subgroups bind 11-*cis* retinal to form UV light-sensitive opsins[7–11]. UV light irradiation of these opsins induces the isomerization of the retinal to the all-*trans* form to produce a visible light-sensitive active state. This active state is thermally stable and photo-converts back to the original dark state. Thus, these opsins have two stable states, the dark and active states, which are inter-convertible with each other by photo-reception. This property is generally observed in bistable opsins[12]. In addition, analysis of the binding preference of retinal isomers revealed that non-mammalian vertebrates Opn5m and Opn5L2 directly bind 11-*cis* and all-*trans* retinals to produce UV light- and visible light-absorbing forms, respectively, whereas mammalian Opn5 exclusively incorporates 11-*cis* retinal to be specialized as a short-wavelength sensor[10]. Recently, analysis of Opn5 knock-out mice revealed that short-wavelength light reception via Opn5 contributes to important physiological functions including the entrainment of the circadian rhythm, the regulation of vascular development and choroidal thickness in the eyes, and the suppression of thermogenesis[13–17]. By contrast, the Opn5L1 subgroup has molecular properties quite different from those of bistable opsin[18]. Opn5L1 exclusively binds all-*trans* retinal to form the active state in the dark. Photoreception by Opn5L1 induces isomerization from all-*trans* to 11-*cis* retinal to suppress the activity. After this process, a covalent adduct is formed between 11-*cis* retinal and the cysteine residue at position 188, which results in conversion of the C11 = C12 double bond to a single bond in the retinal. This leads to thermal rotation of the C11-C12 single bond and the subsequent dissociation of the retinal from the cysteine residue to regenerate the original dark state. That is, Opn5L1 is deactivated by photoreception and spontaneously recovers to the original dark state. Thus, Opn5L1 is defined as photocyclic opsin.

In this study, to reveal the amino acid residue(s) related to the different molecular properties between Opn5m/L2 and Opn5L1 subgroups, we compared the amino acid residues which are predicted to be located around the retinal. The sequence comparison showed that three residues (positions 167, 188, and 212) are different between Opn5m/L2 and Opn5L1[18]. Among the mutant proteins at these positions of Opn5m, T188C mutant Opn5m exclusively bound all-*trans* retinal and acquired the photocyclic property, which is similar to the molecular property of Opn5L1. In addition, our comprehensive mutational analysis of this residue in Opn5m revealed that mutations at position 188 impaired the direct incorporation of 11-*cis* retinal important for forming a UV light-sensitive pigment and led to an incomplete bistable photoreaction. These results show that the residue at

position 188 is crucial for the molecular function of Opn5m as a short-wavelength light-sensitive bistable opsin and serves as a determinant residue that explains quite diverse molecular properties among vertebrates Opn5 subgroups.

## Results

**Search for the amino acid residue(s) required for the binding preference of retinal isomers in Opn5m.** The comparison of the amino acid residues that are located around the retinal in the structure of meta II intermediate of bovine rhodopsin revealed that three residues at positions 167, 188, and 212 (based on the bovine rhodopsin numbering system) are different between Opn5m/L2 and Opn5L1 subgroups (Fig. 1a)[18]. Opn5m subgroup shares tryptophan, threonine, and leucine at positions 167, 188, and 212, whereas Opn5L1 subgroup has phenylalanine/tyrosine, cysteine, and phenylalanine at the corresponding positions (Supplementary Fig. 1). To analyze whether or not these residues are required for the molecular property of Opn5m different from that of Opn5L1, we expressed wild-type and three mutant (W167F, T188C, and L212F) recombinant proteins of *Xenopus tropicalis* Opn5m in cultured cells, because *X. tropicalis* Opn5m showed the highest expression yield among various Opn5m recombinant proteins that we analyzed in a previous study[10]. We reconstituted their photo-pigments by adding 11-*cis* or all-*trans* retinal to the collected cell membranes and purified them by affinity column chromatography using Rho1D4-conjugated agarose (Fig. 1b–e and Supplementary Fig. 2).

Our previous studies showed that wild-type Opn5m protein incorporates 11-*cis* and all-*trans* retinals to produce UV light- and visible light-absorbing forms, respectively. To analyze the mutational effects on the ability to directly bind retinal isomers, we compared the binding preference of 11-*cis* and all-*trans* retinals in wild-type and mutant Opn5m (Fig. 1b and Supplementary Fig. 2a, e). The analysis of the retinal configurations by HPLC showed that wild-type photo-pigment purified after the addition of 11-*cis* retinal exclusively binds 11-*cis* retinal (91%) (upper right panel of Fig. 1b and right panel of Supplementary Fig. 2a), which was confirmed by the absence of a detectable absorption spectral change in the visible region with yellow light irradiation (red curve of the upper left panel in Fig. 1b and left panel in Supplementary Fig. 2a). By contrast, wild-type photo-pigment purified after the addition of all-*trans* retinal had the absorbance in the visible region (black curve of the lower left panel in Fig. 1b) which is derived from the incorporation of all-*trans* retinal (67%) (lower right panel of Fig. 1b and right panel of Supplementary Fig. 2e) and shifted the spectrum into the UV region by yellow light irradiation (red curve of the lower left panel in Fig. 1b and left panel in Supplementary Fig. 2e). In addition, this photo-pigment contained a substantial amount of 11-*cis* retinal (26%) (lower right panel of Fig. 1b). This 11-*cis* retinal is considered to be produced by the isomerization from all-*trans* retinal which is enzymatically catalyzed by intrinsic retinoid processing machinery or non-enzymatically by lipids or nucleophiles present in cell suspensions during the incubation[19–21]. Thus, we concluded that wild-type Opn5m protein preferentially binds 11-*cis* retinal.

Next, we analyzed the retinal configurations of the three mutant recombinant proteins, W167F, T188C, and L212F (Fig. 1c–e and Supplementary Fig. 2b–d, f–h). W167F and L212F mutant proteins purified after the addition of 11-*cis* retinal predominantly bind 11-*cis* retinal (84 and 88%, respectively) (upper right panels of Fig. 1c, d and right panels of Supplementary Fig. 2b, c). This is consistent with the undetectable absorption spectral changes in the visible region with yellow light irradiation in these mutant proteins (red curve

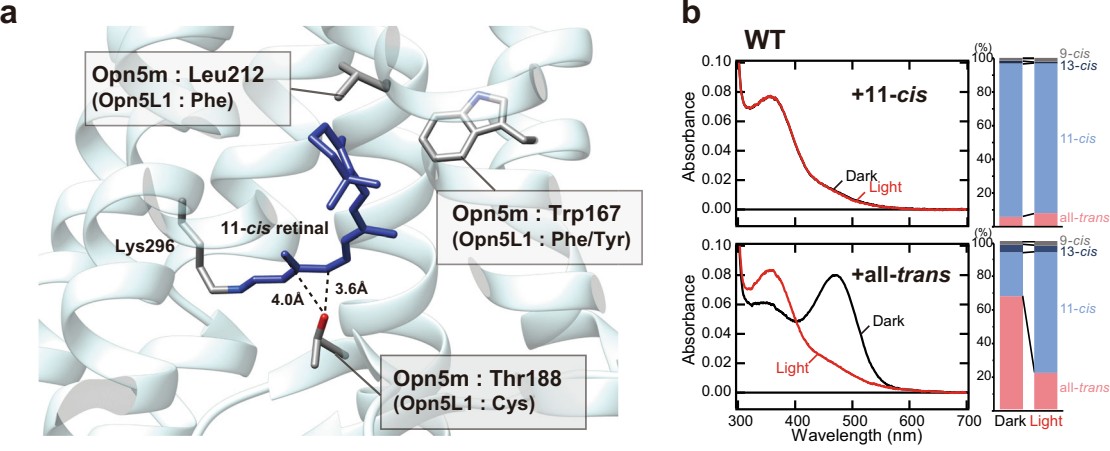

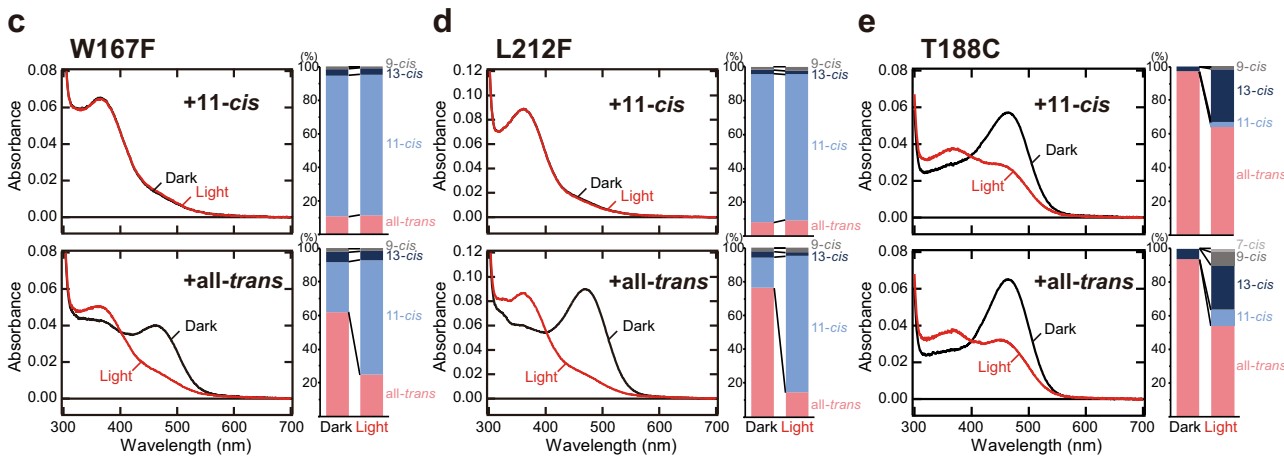

**Fig. 1 Binding selectivity of retinal isomers in wild-type and W167F, L212F, and T188C mutant Opn5m proteins. a** Different amino acid residues around the retinal between Opn5m and Opn5L1 subgroups. A structural model was constructed based on the crystal structure of squid rhodopsin (PDB: 2z73) visualized using UCSF Chimera[27]. Squid rhodopsin has serine and phenylalanine at positions 188 and 212 (based on the bovine rhodopsin numbering system) (Supplementary Fig. 1), which were replaced by threonine and leucine in this model. **b-e** (left) Absorption spectra of Opn5m wild-type (**b**) and W167F (**c**), L212F (**d**), and T188C (**e**) mutant proteins purified after the addition of 11-*cis* or all-*trans* retinal to the collected cell membranes. The spectra were recorded at 0 °C in the dark (curve 1) and after yellow light (>500 nm) irradiation (curve 2). Spectral change by light irradiation is shown in left panels of Supplementary Fig. 2a–h. (right) Retinal configuration changes of Opn5m wild-type (**b**) and W167F (**c**), L212F (**d**), and T188C (**e**) mutant proteins purified after the addition of 11-*cis* or all-*trans* retinal to the collected cell membranes. The retinal isomers before and after yellow light (>500 nm) irradiation were analyzed with HPLC after extraction of the chromophore as retinal oximes (right panels of Supplementary Fig. 2a–h). The negative peak in the visible region of the difference spectrum calculated before and after yellow light irradiation of the wild-type (Supplementary Fig. 2e) corresponds to λmax of the all-*trans* retinal bound form (474 nm). λmax of the all-*trans* retinal bound form of the mutant proteins calculated from their difference spectra (Supplementary Fig. 2f–h) are 470 nm (W167F), 475 nm (L212F), and 470 nm (T188C).

of the upper left panels in Fig. 1c, d and left panels in Supplementary Fig. 2b, c). Moreover, these mutant proteins purified after the addition of all-*trans* retinal had the absorbance in the visible region and shifted the spectra to the UV region by yellow light irradiation (black and red curves of lower left panels in Fig. 1c, d and left panels in Supplementary Fig. 2f, g). These samples also contained a substantial amount of 11-*cis* retinal (30 and 18%, respectively) (lower right panels of Fig. 1c, d and right panels of Supplementary Fig. 2f, g). Thus, the binding preference of the retinal isomers and the spectral change of W167F and L212F mutant proteins were quite similar to those of wild-type. On the other hand, T188C mutant protein purified after the addition of 11-*cis* retinal had the absorption maximum (λmax) in the visible region (black curve of the upper left panel in Fig. 1e). This sample contained a predominant amount of all-*trans* retinal (97%) with an undetectable amount of 11-*cis* retinal

(upper right panel of Fig. 1e and right panel of Supplementary Fig. 2d). The presence of all-*trans* retinal bound form was confirmed by the yellow light-dependent spectral shift to the UV region (red curve of the upper left panel in Fig. 1e and left panel in Supplementary Fig. 2d), which was not observed in wild-type and two other mutant proteins (W167F and L212F). T188C mutant protein purified after the addition of all-*trans* retinal also contained predominantly all-*trans* retinal (94%) (lower right panel of Fig. 1e and right panel of Supplementary Fig. 2h) and showed the absorption spectrum (black curve of the lower left panel in Fig. 1e) quite similar to that of the sample prepared after the addition of 11-*cis* retinal. Yellow light irradiation of this sample induced the *trans-cis* isomerization of the retinal (lower right panel of Fig. 1e) and shifted the absorption spectra into the UV region (red curve of the lower left panel in Fig. 1e and left panel in Supplementary

Fig. 2h). These results showed that T188C mutant protein of Opn5m preferentially binds all-*trans* retinal compared to wild-type. This was similar to the property of Opn5L1 which loses the ability to directly incorporate 11-*cis* retinal and exclusively binds all-*trans* retinal. Therefore, the well-conserved Thr188 in the Opn5m subgroup is likely to be required for the binding preference of retinal isomers.

**Binding preference of retinal isomers in Thr188 mutant Opn5m.** To reveal the roles of the well-conserved threonine residue at position 188 in the Opn5m subgroup, we replaced Thr188 with the other 18 natural amino acid residues in addition to the cysteine of the T188C mutant protein. We expressed the mutant recombinant proteins in cultured cells and reconstituted the photo-pigments by adding 11-*cis* or all-*trans* retinal to the collected cell membranes. Among these mutant Opn5m, we could produce the photo-pigments of six mutant proteins, T188S, T188G, T188A, T188N, T188M, and T188Q (Fig. 2 and Supplementary Figs. 3, 4), but could not detect the formation of the photo-pigments of the other 12 mutant proteins by absorption spectrum analysis. This showed that introduction of amino acid residues which have positively charged (Arg, His, and Lys), negatively charged (Asp and Glu), bulky aromatic (Phe, Trp, and Tyr), cyclic (Pro), or branched (Ile, Leu, and Val) side chains into this position prevented the formation of the photo-pigments of Opn5m.

To analyze whether or not Thr188 regulates the ability to directly bind retinal isomers, we compared the binding preference of 11-*cis* and all-*trans* retinals in wild-type and mutant Opn5m by analysis of the retinal configurations of the mutant proteins (Fig. 2 and Supplementary Figs. 3, 4). All the mutant proteins purified after the addition of all-*trans* retinal also had the absorbance in the visible region (black curve of Fig. 2g–l). These samples contained a smaller amount of 11-*cis* retinal (0–5%) than wild-type (26%, shown in Fig. 1b) in addition to a large amount of all-*trans* retinal (71–93%) (right panels of Fig. 2g–l and Supplementary Fig. 4g–l). Yellow light irradiation of these samples induced the *trans-cis* isomerization of the retinal (right panels of Fig. 2g–l) and shifted the absorption spectra into the UV region (red curve of Fig. 2g–l and Supplementary Fig. 3g–l). Moreover, all the mutant proteins purified after the addition of 11-*cis* retinal contained a larger amount of all-*trans* retinal (more than 17%) than wild-type (5%, shown in Fig. 1b) (right panels of Fig. 2a–f and Supplementary Fig. 4a–f). In particular, T188N and T188M mutant proteins contained predominantly all-*trans* retinal (85 and 86%, respectively) and a small amount of 11-*cis* retinal (3 and 2%, respectively) (Fig. 2d, e). The presence of the all-*trans* retinal bound form was confirmed by their yellow light-dependent spectral shift to the UV region (red curve of Fig. 2a–f and Supplementary Fig. 3a–f), which was not observed in wild-type (Fig. 1b). These results showed that all the mutant Opn5m preferentially bind all-*trans* retinal compared to wild-type. Thus, the well-conserved Thr188 in the Opn5m subgroup is required for the maintenance of the ability to directly bind 11-*cis* retinal, which enables Opn5m to efficiently work as a short-wavelength sensor.

**Spectral changes of Thr188 mutant Opn5m.** Next, we analyzed the spectral properties of the mutant proteins. As shown in Figs. 1 and 2, several Opn5m Thr188 mutant proteins lost the ability to directly bind 11-*cis* retinal. Therefore, we expressed the recombinant proteins of the seven mutant Opn5m in cultured cells and purified them after reconstitution of the photo-pigments by the addition of all-*trans* retinal to the medium. The wild-type and mutant proteins had the absorption peak in the visible region, which was shifted to the UV region by yellow light irradiation

(Fig. 3). The negative peak in the visible region of the difference spectrum calculated before and after yellow light irradiation of the wild-type corresponds to λmax of the all-*trans* retinal bound form (474 nm) (Supplementary Fig. 5a). We also calculated λmax of the all-*trans* retinal bound form of the mutant proteins based on their difference spectra (Supplementary Fig. 5b–h). λmax of T188N (460 nm), T188M (467 nm), and T188C (470 nm) were blue-shifted and those of T188S (478 nm), T188G (477 nm), T188A (481 nm), and T188Q (483 nm) were red-shifted compared to that of wild-type.

Moreover, we compared the detailed spectral changes between wild-type and mutant proteins of Opn5m. In wild-type, yellow light irradiation shifted the spectra to the UV region, and subsequent UV and yellow light irradiations repeatedly induced a simultaneous increase and decrease of the absorption in the visible and UV region (Fig. 3a). Also in T188S, T188G, and T188A mutant proteins, alternating irradiation by yellow light and UV light led to spectral changes in the visible and UV regions (Fig. 3b–d). However, the spectral changes induced by the second yellow and UV light irradiations were smaller than those induced by the first light irradiations (curves 3 and 5 of Fig. 3b–d). Thus, T188S, T188G, and T188A exhibited a partially incomplete light-dependent inter-convertibility of two stable forms. T188N and T188M mutant proteins shifted their absorption spectra into the UV region upon yellow light irradiation (curve 2 of Fig. 3e, f). However, subsequent UV light irradiation resulted in a quite small conversion from a UV light-absorbing form to a visible light-absorbing form (curve 3 of Fig. 3e, f). These results show that T188N and T188M markedly weakened the ability to recover a visible light-absorbing form in a light-dependent manner. T188Q and T188C mutant proteins were also converted to a visible light-absorbing form by yellow light irradiation (curve 2 of Fig. 3g, h). However, subsequent UV and yellow light irradiations triggered gradually attenuated photoreactions (curves 3–6 of Fig. 3g, h). Thus, T188Q and T188C mutations impaired the light-dependent inter-convertibility of two stable forms. Altogether, the systematic mutational analysis revealed that the well-conserved Thr188 in the Opn5m subgroup is crucial for a complete inter-convertibility of two stable forms by light irradiations.

**Acquisition of photocyclic property in T188C mutant Opn5m.** Our previous study showed that the cysteine residue at position 188 is conserved in the Opn5L1 subgroup and has a crucial role in the photocyclic property. Thus, we analyzed whether or not T188C mutation of Opn5m alters the molecular property to mimic that of Opn5L1 (Fig. 4 and Supplementary Fig. 6). Yellow light irradiation of T188C mutant protein decreased the absorption in the visible region and increased the absorption in the UV region at 10 °C (curve 2 of Fig. 4a). Subsequently, the absorption in the visible region recovered without additional light irradiation (curves 3–6 of Fig. 4a). This is in contrast with the observation that wild-type and T188A mutant proteins showed no detectable spectral changes during the incubation after yellow light irradiation (Supplementary Fig. 6c–e). Our time-resolved spectro-photometric experiment with T188C mutant protein at 37 °C confirmed that the absorption in the visible region was decreased by flash light and spontaneously recovered (curves 2–6 of Fig. 4b). This recovery process of the mutant protein at 37 °C was well-fitted with double-exponential functions (Fig. 4c). To assess the molecular mechanism of the photocyclic property of T188C mutant protein, we analyzed the change of the retinal configuration in the reaction process (Fig. 4d). Yellow light irradiation of T188C mutant protein produced a large amount of 13-*cis* retinal (45%) and a small amount of 11-*cis* retinal (11%). After

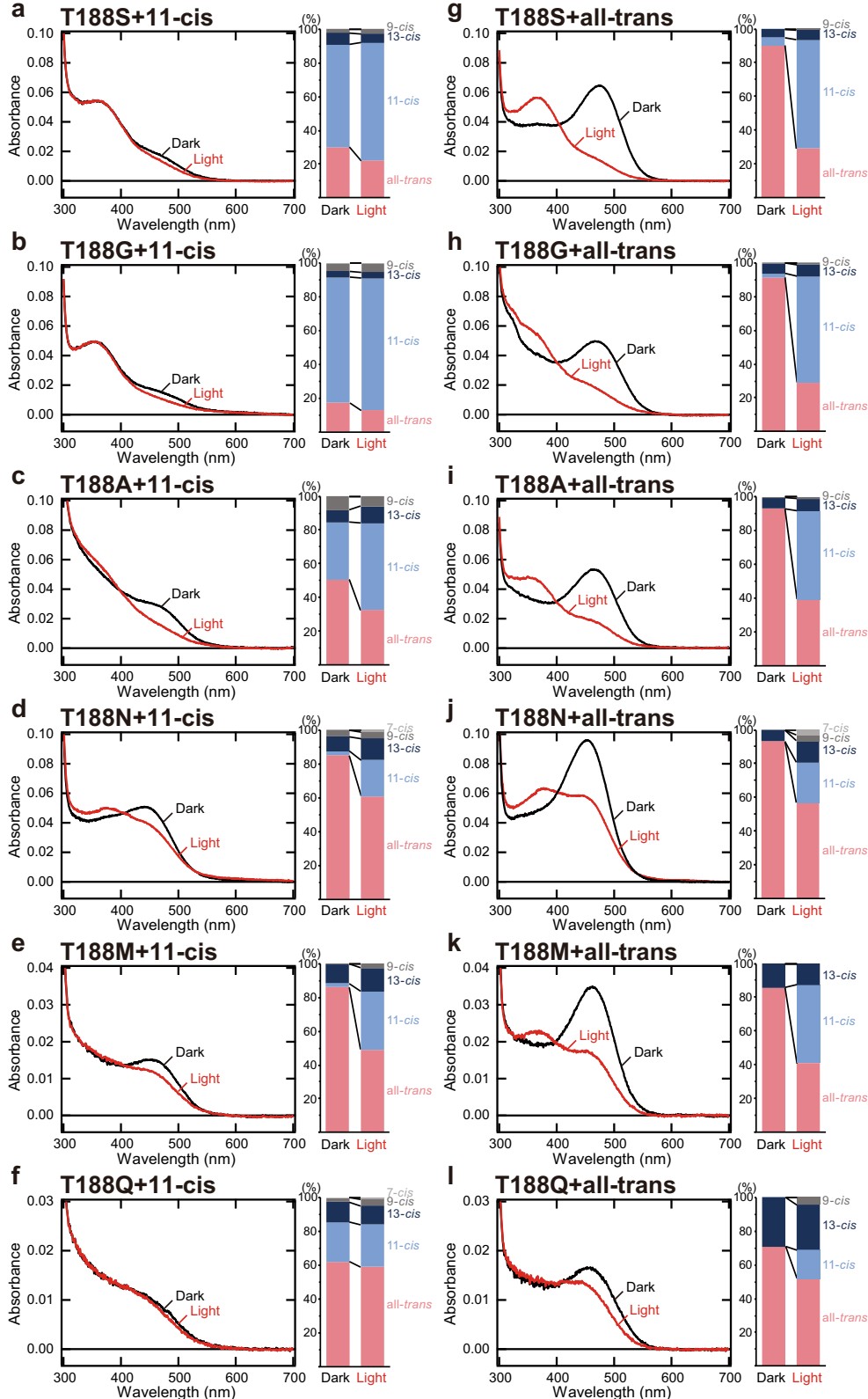

the incubation in the dark, we observed the re-isomerization of the retinal from 13-*cis* or 11-*cis* form to all-*trans* form to induce the thermal recovery of the absorbance in the visible region. The double-exponential fitting of the recovery process at 37 °C can be explained by the difference in the recovery rate of all-*trans* retinal from 11-*cis* and 13-*cis* retinals (Supplementary

Fig. 6f). Thus, T188C mutant Opn5m can self-regenerate to the original dark state after light irradiation by the combination of the photoisomerization and thermal isomerization of the retinal. Altogether, the T188C mutation of Opn5m drastically altered the binding preference of the retinal isomers and the photoreaction property to partially mimic those of Opn5L1.

**Fig. 2 Binding selectivity of retinal isomers in Opn5m Thr188 mutant proteins. a–f** (left) Absorption spectra of Opn5m T188S (**a**), T188G (**b**), T188A (**c**), T188N (**d**), T188M (**e**), and T188Q (**f**) mutant proteins purified after the addition of 11-*cis* retinal to the collected cell membranes. The spectra were recorded at 0 °C in the dark (curve 1) and after yellow light irradiation (curve 2). Spectral change by light irradiation is shown in Supplementary Fig. 3. (right) Retinal configuration changes of Opn5m T188S (**a**), T188G (**b**), T188A (**c**), T188N (**d**), T188M (**e**), and T188Q (**f**) mutant proteins purified after the addition of 11-*cis* retinal to the collected cell membranes. The retinal isomers before and after yellow light (>500 nm) irradiation were analyzed with HPLC after extraction of the chromophore as retinal oximes. (Supplementary Fig. 4a–f). **g–l** (left) Absorption spectra of Opn5m T188S (**g**), T188G (**h**), T188A (**i**), T188N (**j**), T188M (**k**), and T188Q (**l**) mutant proteins purified after the addition of all-*trans* retinal to the collected cell membranes. The spectra were recorded at 0 °C in the dark (curve 1) and after yellow light irradiation (curve 2). Spectral change by light irradiation is shown in Supplementary Fig. 3. (right) Retinal configuration changes of Opn5m T188S (**g**), T188G (**h**) T188A (**i**), T188N (**j**), T188M (**k**), and T188Q (**l**) mutant proteins purified after the addition of all-*trans* retinal to the collected cell membranes. The retinal isomers before and after yellow light (>500 nm) irradiation were analyzed with HPLC after extraction of the chromophore as retinal oximes. (Supplementary Fig. 4g–l).

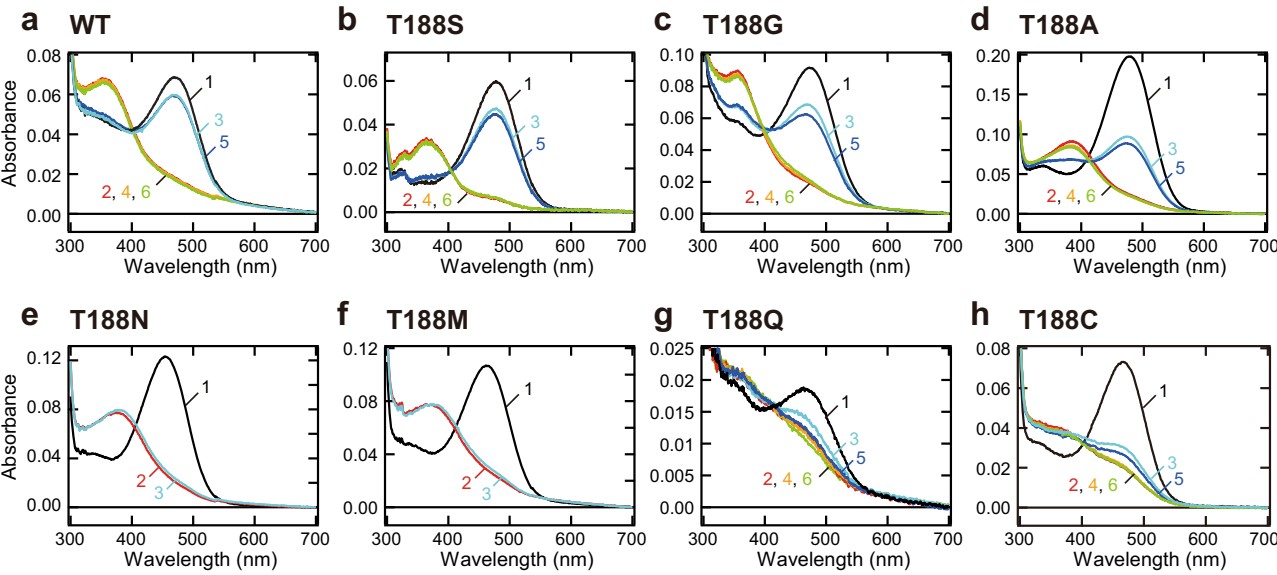

**Fig. 3 Spectral changes of wild-type and Thr188 mutant Opn5m proteins.** Absorption spectra of Opn5m wild-type (**a**) and T188S (**b**), T188G (**c**), T188A (**d**), T188N (**e**), T188M (**f**), T188Q (**g**), and T188C (**h**) mutant proteins purified after the addition of all-*trans* retinal to the medium of the transfected cultured cells. The spectra were recorded at 0 °C in the dark (curve 1), after yellow light (>500 nm) irradiation (curve 2), after subsequent UV light (360 nm) irradiation (curve 3), after yellow light re-irradiation (curve 4), after UV light re-irradiation (curve 5) and after yellow light re-irradiation (curve 6). Spectral changes by each light irradiation are shown in Supplementary Fig. 5.

## Discussion

In this study, to assess the role of the well-conserved threonine residue at position 188 in the Opn5m subgroup, we replaced Thr188 of Opn5m with the other 19 natural amino acid residues and prepared these recombinant proteins in cultured cells. However, we were not able to detect the formation of the photopigments of 12 mutant proteins among them. Thus, the amino acid residues which have positively charged (Arg, His, and Lys), negatively charged (Asp and Glu), bulky aromatic (Phe, Trp, and Tyr), cyclic (Pro), or branched (Ile, Leu, and Val) side chains were not suitable for this position, which is consistent with the failure of human rhodopsin G188E and G188R mutant proteins to form photo-pigments[22]. This suggested that the space available around this position of Opn5m is limited and hydrophobic. Then we successfully obtained the photo-pigments of seven mutant proteins after reconstitution with retinal. Absorption spectra of the recombinant proteins of mutant Opn5m revealed that all the mutations shifted the λmax of the all-*trans* retinal bound form. In addition, all the mutations impaired the ability to directly bind 11-*cis* retinal and the light-dependent inter-convertibility of 11-*cis* retinal and all-*trans* retinal bound forms. Thus, Thr188 in Opn5m is essential for efficient incorporation of 11-*cis* retinal to form a bistable short-wavelength sensor. The crystal structure of squid rhodopsin, which is a prototypical bistable opsin, shows that the residue at position 188 is accommodated in a β-sheet structure of the second

extracellular loop and is located within 5 Å of the retinal polyene chain (Fig. 1a)[23]. Thus, the mutations at position 188 of Opn5m may perturb the local environment that connects the retinal and the second extracellular loop, which in turn induces a spectral shift and hampers the direct binding of 11-*cis* retinal and the efficient *cis*-*trans* photoisomerization of the retinal. We previously reported that mammalian Opn5m protein evolved to lose the ability to directly bind all-*trans* retinal by single amino acid replacement at position 168[10]. That finding taken together with the present result indicates that quite a few amino acid residues can strongly regulate the binding selectivity of the retinal isomers in Opn5m.

Moreover, the present study revealed that the T188S mutant protein maintains the ability to directly bind 11-*cis* retinal and the bistable photoreaction relatively well. Opn5L2 contains a threonine or serine residue at position 188 and has the ability to directly incorporate both 11-*cis* and all-*trans* retinals and to undergo light-dependent inter-conversion between the 11-*cis* and all-*trans* retinal bound forms. Thus, we speculate that replacement with a serine residue at position 188 is relatively acceptable in the Opn5L2 subgroup. Comparison of the amino acid sequences among various opsins also shows that bistable opsins generally have a threonine or serine residue at position 188 (Supplementary Fig. 7). Thus, the residue at position 188 may be required for the bistable photoreaction and the binding preference of retinal isomers in other bistable opsins.

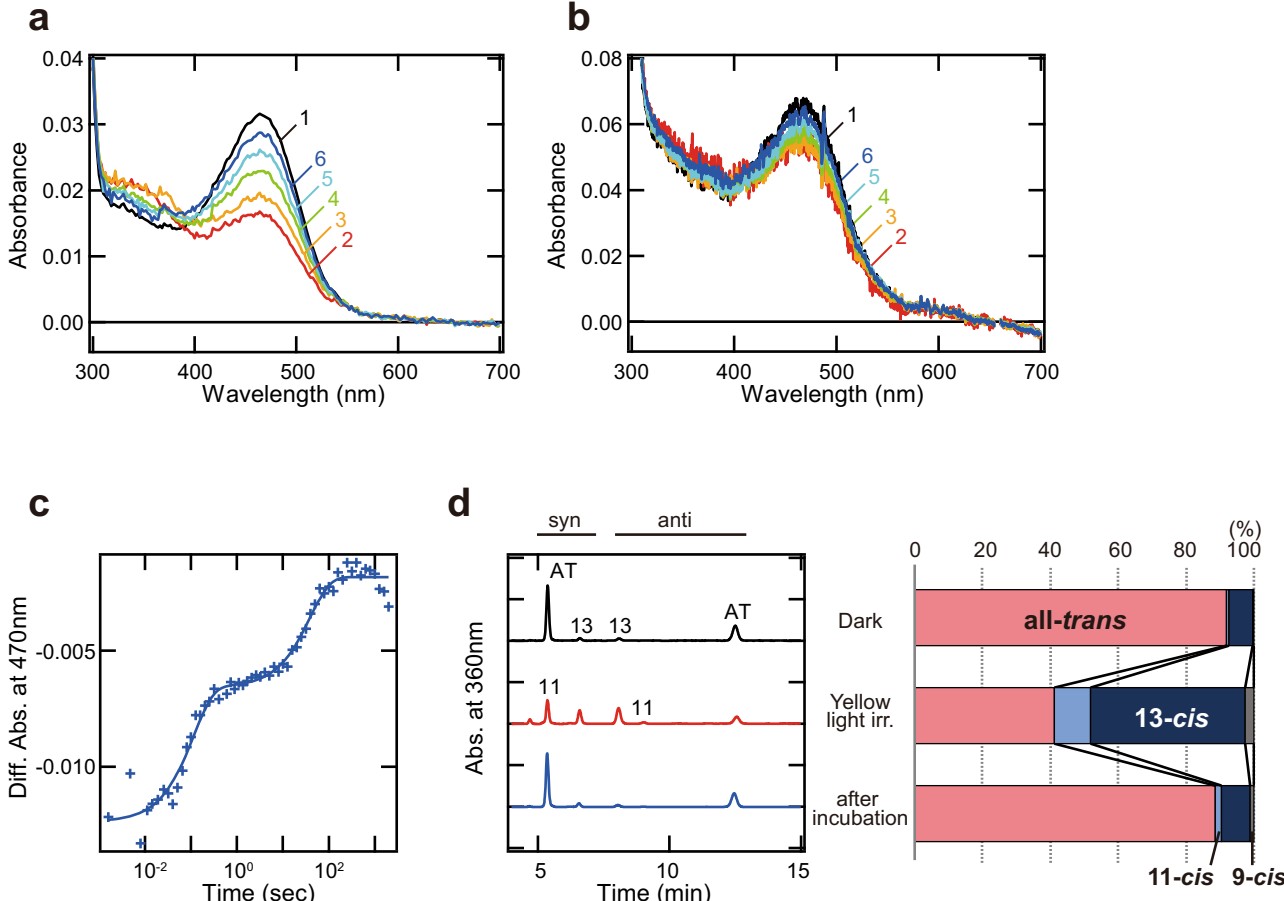

**Fig. 4 Molecular characteristics of T188C mutant Opn5m protein. a, b** Absorption spectra of Opn5m T188C mutant protein purified after the addition of all-*trans* retinal to the medium of the transfected cultured cells. The spectra were recorded at 10 °C (**a**) in the dark (curve 1) and 0, 4.33, 16.5, 37.5, and 116 min after yellow light (>500 nm) irradiation (curve 2–6). Spectral changes observed during the incubation in the dark after light irradiation are shown in Supplementary Fig. 6a. The spectra of T188C mutant protein were also recorded at 37 °C (**b**) in the dark (curve 1) and 0.0016, 0.032, 0.10, 3.26, 1995 s after yellow flash light (>460 nm) irradiation (curve 2–6). Spectral changes observed during the incubation in the dark after light irradiation are shown in Supplementary Fig. 6b. **c** Absorption change of Opn5m T188C mutant protein at 470 nm after light irradiation at 37 °C. Time course of the absorbance at 470 nm was plotted according to the data in Supplementary Fig. 6b and was fitted with a double-exponential function ($\tau = 0.11$ and 39.2 s). These time constants are more than 200 times faster than that of chicken Opn5L1 ($\tau = 175$ min) as described previously[18]. **d** Retinal configuration analysis of Opn5m T188C mutant protein. (Left) Retinal isomers of Opn5m T188C mutant protein purified after reconstitution with all-*trans* retinal were analyzed with HPLC after extraction of the chromophore from the samples before irradiation (black), after yellow light irradiation (red) and after subsequent incubation for 2 h in the dark (blue) as retinal oximes. AT, all-*trans* retinal; 11, 11-*cis* retinal; 13, 13-*cis* retinal. (Right) Isomeric compositions of retinal before light irradiation, after light irradiation, and after subsequent incubation.

It is notable that T188C mutant protein exclusively bound all-*trans* retinal and acquired photocyclic properties similar to those of Opn5L1. Our previous report provided a molecular model of Opn5L1 function, that is, Opn5L1 forms an adduct between Cys188 and the retinal after photoreception, which results in the conversion of the C11=C12 double bond to a single bond in the retinal and accelerates the thermal recovery to the original dark state[18]. During this photocyclic process of chicken Opn5L1, we observed an increase of the absorbance at 270 nm, which is indicative of the breakdown of the retinal-conjugated double bond system due to the adduct formation. Thus, we speculate that this molecular model is applicable to T188C mutant Opn5m. However, T188C mutant Opn5m had a photocycle rate (Fig. 4c) more than 200 times faster than that previously found for chicken Opn5L1 and showed an increase of the absorbance at around 380 nm after photoreception, instead of detectable changes of the absorbance at around 270 nm (Supplementary Fig. 6a, b). This is probably because the intermediate of Opn5m T188C mutant protein that contains

the adduct between the introduced cysteine residue and the retinal has too short a lifetime to detect. In addition, our analysis revealed that Opn5m T188C mutant protein conducts the photocycle via photo-conversion from all-*trans* retinal to not only 11-*cis* but also 13-*cis* retinal (Supplementary Fig. 6f), whereas the photocycle of Opn5L1 is triggered by the photo-conversion from all-*trans* retinal to 11-*cis* retinal. Therefore, it is suggested that the structural differences in the retinal binding pocket between Opn5L1 and T188C mutant Opn5m can alter the rate of the photocycle and the selectivity of the retinal isomerization. Additional mutations in Opn5m would be necessary to completely mimic the molecular properties of Opn5L1 by recapitulating the molecular evolutionary process from bistable opsin to photocyclic opsin.

In conclusion, we revealed that Thr188 is well conserved in the Opn5m subgroup and contributes to the efficient incorporation of 11-*cis* retinal and the maintenance of the bistable photoreaction in Opn5m. Moreover, the T188C mutation of Opn5m led to the acquisition of exclusive incorporation of all-*trans* retinal and

photocyclic properties similar to those of Opn5L1. Therefore, the residue at position 188 is a determinant residue of quite diverse molecular properties among vertebrate Opn5 subgroups. Further analysis of the detailed molecular mechanism by which the single residue at position 188 can drastically regulate the photoreaction profile and the binding selectivity of retinal isomers will provide valuable insights into the diversification of the molecular properties of various opsins.

## Methods

**Preparation of Opn5 recombinant proteins**. To improve the expression level of *X. tropicalis* Opn5m (GenBank accession number XM_002935990) recombinant protein, we truncated 21 amino acid residues from the C-terminus, as mentioned in our previous paper[10]. The C-terminal truncated cDNA of *X. tropicalis* Opn5m was tagged with the epitope sequence of the anti-bovine rhodopsin monoclonal antibody Rho1D4 at the C-terminus and was introduced into the mammalian expression vector pCAGGS[24]. Site-directed mutations were introduced using an In-Fusion cloning kit (Clontech) according to the manufacturer's instructions. The plasmid DNA was transfected into HEK293S cells using the calcium phosphate method. To obtain the Opn5m mutant recombinant proteins for the analysis of their spectral changes (Figs. 3, 4), 5 µM all-*trans* retinal was added to the medium 24 h after transfection and the cells were kept in the dark before the collection of the cells 48 h after transfection. To obtain the recombinant proteins for the analysis of the preference of retinal isomers (Figs. 1, 2), the cells were collected 48 h after transfection without the addition of retinal to the medium. The collected cells were incubated with 40 µM 11-*cis* or all-*trans* retinal at 4 °C for 24 h in the dark. The reconstituted pigments were extracted from cell membranes with 1% dodecyl maltoside (DDM) in Buffer A (50 mM HEPES, 140 mM NaCl, pH 6.5) and were purified using Rho1D4-conjugated agarose. The purified pigments were eluted with 0.02% DDM in Buffer A containing the synthetic peptide that corresponds to the C-terminus of bovine rhodopsin. All the procedures were carried out on ice under dim red light.

**Spectrophotometry**. UV/Vis absorption spectra were recorded using a Shimadzu UV2400 or UV2450 spectrophotometer and an optical cell (width, 2 mm; light path, 1 cm). The sample temperature was maintained by a temperature controller (RTE-210, NESLAB) at 0 ± 0.1 °C. The sample was irradiated with light which was generated by a 1-kW tungsten halogen lamp (Master HILUX-HR, Rikagaku Seiki) and passed through optical filters (Y-52 or UV-D36C, AGC Techno Glass). The transient absorption spectra of Opn5m T188C mutant protein were recorded using a Shimadzu UV2450 spectrophotometer at 10 ± 0.1 °C. The spectra were measured in the dark and after 2 min irradiation with light which was generated by a 1-kW tungsten halogen lamp and passed through an optical filter (Y-52, AGC Techno Glass). The transient absorption spectra of Opn5m T188C mutant protein were also recorded using a high-speed CCD camera spectrophotometer (C10000 system, Hamamatsu Photonics) kept at 37 ± 0.1 °C by a temperature controller (qpod, QUANTUM Northwest)[25] to accelerate the photocyclic reaction, according to the methods used in our previous study for Opn5L1[18]. The flash light for irradiation was generated by a short-arc power flash (SA-200, Nissin Electronic; pulse duration of ~170 µs and flash lamp input of 200 J/F) and passed through an optical filter (Y-48, AGC Techno Glass).

**HPLC analysis of retinal isomers**. The retinal configurations of samples were analyzed by HPLC (LC-10ATvp, Shimadzu) with a silica column (YMC-Pack SIL, particle size 3 µm, 150 × 6.0 mm, YMC)[26]. The solvent composition was 98.8% (v/v) benzene, 1.0% (v/v) diethyl ether, and 0.2% (v/v) 2-propanol. The aliquots of purified Opn5m proteins kept in the dark or irradiated with light were diluted to 250 µL and treated with 25 µL of 1 M hydroxylamine. After 1 h incubation, the sample was mixed with 250 µL methanol and 250 µL dichloromethane to extract retinal oximes. The retinal oximes were isolated by phase separation with 1 mL of hexane. The hexane layer was collected and dried with 0.5 g anhydrous Na₂SO₃. This procedure was repeated twice. The collected 2 mL hexane was evaporated by N₂ gas. The dried sample was dissolved in 30 µL hexane, and 10 µL of it was used for the HPLC analysis.

**Statistics and reproducibility**. Statistical tests were not performed in this study. The spectral experiments were repeated at least twice with similar results and representative results are shown.

**Reporting summary**. Further information on research design is available in the Nature Research Reporting Summary linked to this article.

## Data availability
The source data underlying Figs. 1, 2, 3 and 4 are provided as Supplementary Data 1, 2, 3 and 4, respectively. Any remaining information can be obtained from the corresponding author upon reasonable request.

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

## Acknowledgements
We thank Dr. E. Nakajima for critical reading of the manuscript. We also thank Prof. R. S. Molday for the generous gift of a Rho1D4-producing hybridoma and Prof. J. Nathans for providing the HEK293S cell line. This work was supported in part by Grants-in-Aid for Scientific Research of MEXT to KS (20K08885), YI (19K21848), YS (16H02515), and TY (16K07437), CREST, JST JPMJCR1753 (TY), a grant from the Kyoto University Foundation (TY), a grant from the Takeda Science Foundation (TY), and a grant from Fujiwara Natural History Foundation (CF).

## Author contributions

C.F., K.S., Y.S. and T.Y. designed the research. C.F., K.S., Y.N. and T.Y. conducted the experiments. C.F., K.S., Y.N., Y.I., H.O., Y.S. and T.Y. analyzed the data. C.F., K.S., Y.S. and T.Y. wrote the manuscript with editing by all authors.

## Competing interests

The authors declare no competing interests.
