## [Peer Review File · Communications Biology]

Reviewers' comments:

Reviewer #1 (Remarks to the Author):

In this manuscript, the authors analyzed 7 mutants of Opn5m from *Xenopus tropicalis* (T188S, T188G, T188A, T188N, T188M, T188Q, and T188C) and found that T188 is involved in the bistable property. The authors provide new spectroscopic aspects of Opn5m from *Xenopus tropicalis*. However, as a whole, the manuscript does not seem to have much novelty.

The title of the paper is "Identification of the amino acid residue responsible for UV-sensitive bistable property of vertebrate non-visual opsin Opn5", but the authors only analyzed the single amino acid (T188) of Opn5 from the single species (*Xenopus tropicalis*). The authors have to perform more comprehensive analysis, including the analysis of other residues and other Opn5 homologs, otherwise, the scope of the manuscript is too specific for the broad audience of Communications Biology.

One of the major claims in the manuscript is that the T188C mutation of Opn5m converted the photoreaction profile and the binding selectivity of retinal isomers to mimic those of Opn5L1. In Opn5L1, it is reported that (1) the light illumination photo-isomerizes all-trans-retinal to 11-cis, and (2) the retinal transiently forms covalent interaction with C188 (Sato et al., 2018). However, in the T188C mutant of Opn5m, the retinal is isomerized to 13-cis (Fig. 2d,h,l), and the authors do not provide any direct evidence of the adduct formation between T188C and retinal. The mutation surely affects the bistable property of Opn5m, but it is too much to claim that the mutant mimics the property of Opn5L1.

Minor points

1. Page 10, lines 3-6: "The crystal structure of squid rhodopsin, which is a prototypical bistable opsin, shows that the residue at position 188 is accommodated in a β -sheet structure of the second extracellular loop and is located within 5 Å of the retinal polyene chain (Fig. S5a)." How close are the sequences of squid rhodopsin and Opn5m? The authors should at least show the sequence alignment between squid rhodopsin and Opn5m, and they can also show the homology model of Opn5m based on the crystal structure of squid rhodopsin.

2. The authors show the results of "Binding selectivity of retinal isomers" separately in Fig. 1 and 2, but this is very difficult to see. The authors can merge Fig. 1 and 2, keep action spectra to the main figure, and move the bar graphs to supplementary.

Reviewer #2 (Remarks to the Author):

This is an interesting study that confirms the role of Thr188 in the functionality of Opn5m. The account however is very difficult to follow, partly through grammatical errors, partly through being too condensed and partly because the Figures 1 and 2 which are at the centre of the investigation, are difficult to interpret. It is imperative that the experiments are much more extensively explained, and the rationale involved, in the Results section.

Here are my major comments

Firstly, why no page or line numbering?

Results page 5: The reader needs to know what is meant by "all possible 19 mutations".

Coding/non-coding changes?

1. Change to read "wild type and mutant recombinant proteins"
2. What does this sentence mean? "Among these mutants, we detected the photopigments of seven mutants...." Only these mutant proteins produced photopigments?
3. Say what form each retinal produces.
4. Change to read "wild-type and mutant Opn5m". the grammatical structure used here and elsewhere in the text is very poor.
5. Define what is meant by "purified"

Page 6

6. Need to state more clearly the illumination used in the experiments.
7. Is it a "speculation" or a "conclusion"?
8. Avoid using "mutants" – it is mutant protein of mutant Opn5m
9. The various experiments presented in the Figs need to be described in the Results –it is not sufficient to expect the reader to extract this from Fig legends and M&M.
10. "Larger amount" – meaning what? Give a range if single figure not possible.
11. "More preferentially" – bad grammar – either preferentially or not!
12. "is responsible" – to say "is required". Other sites may also be involve so to say that this is all down to one site may be going too far.

Page 7

13. Use of "mutants" again.

Page 8

14. Significance of temperature not explained.

Page 9

15. "all possible 19 mutations" – define
16. Move at least some of the description of the effect of the different mutations into the Results section.

Fig 1 is very difficult to follow. I presume the 1 and 2 labelling of spectra refer to 11-cis and all trans. Suggest that the mutation is put on Figure alongside a – d. Bar charts may be easier to understand, but need much more description in Results section.

Figs 1 and 2 need to be combined into single figure.

Figure 3 – put wild-type and mutations on Figure, not just in legend.

Responses to Reviewer #1

Thank you very much for your critical reading of our manuscript and for your valuable comments. We revised the manuscript based on your comments. The following are our responses to your comments.

Comment 1: The title of the paper is “Identification of the amino acid residue responsible for UV-sensitive bistable property of vertebrate non-visual opsin Opn5”, but the authors only analyzed the single amino acid (T188) of Opn5 from the single species (*Xenopus tropicalis*). The authors have to perform more comprehensive analysis, including the analysis of other residues and other Opn5 homologs, otherwise, the scope of the manuscript is too specific for the broad audience of *Communications Biology*.

Response: According to your comment, we performed additional experiments. Opn5m and Opn5L2 share molecular properties as UV light-sensitive bistable opsins (Yamashita et al., (2010) PNAS; Ohuchi et al., (2012) PLoS One; Yamashita et al., (2014) J. Biol. Chem.; Sato et al., (2016) PLoS One). By contrast, Opn5L1 has quite different molecular properties, that is, deactivation by visible light reception and subsequent recovery to the dark state by photocyclic reaction (Sato et al., (2018) Nat. Commun.). To identify the amino acid residue(s) related to the different molecular properties between Opn5m/L2 and Opn5L1 subgroups, we compared the amino acid residues which are predicted to be located around the retinal. The sequence comparison showed that three residues (positions 167, 188 and 212) are different between Opn5m/L2 and Opn5L1 (Sato et al., (2018) Nat. Commun.). The Opn5m subgroup shares tryptophan, threonine and leucine at positions 167, 188 and 212, whereas the

Opn5L1 subgroup has phenylalanine/tyrosine, cysteine and phenylalanine at the corresponding positions. Among the mutant proteins at these positions of Opn5m, W167F and L212F mutant Opn5m had molecular properties quite similar to those of wild-type. By contrast, T188C mutant Opn5m exclusively bound all-trans retinal and acquired the photocyclic property, which is relatively similar to the molecular properties of Opn5L1. This evidence supports the importance of Thr188 in Opn5m. We also tried to prepare T188C mutant proteins of Opn5m and Opn5L2 from another species. Although we had already analyzed in detail the molecular properties of chicken Opn5m and Opn5L2 (Yamashita et al., (2010) PNAS; Ohuchi et al., (2012) PLoS One), we could not detect expression of the recombinant T188C mutant proteins of chicken Opn5m and Opn5L2 in cultured cells. This is probably because *X. tropicalis* Opn5m shows the highest expression yield among various Opn5m and Opn5L2 recombinant proteins that we analyzed. Thus, we added the results of W167F and L212F mutant Opn5m in Fig. 1 and substantially revised the manuscript including the title.

Comment 2: One of the major claims in the manuscript is that the T188C mutation of Opn5m converted the photoreaction profile and the binding selectivity of retinal isomers to mimic those of Opn5L1. In Opn5L1, it is reported that (1) the light illumination photo-isomerizes all-trans-retinal to 11-cis, and (2) the retinal transiently forms covalent interaction with C188 (Sato et al., 2018). However, in the T188C mutant of Opn5m, the retinal is isomerized to 13-cis (Fig. 2d,h,l), and the authors do not provide any direct evidence of the adduct formation between T188C and retinal. The mutation surely affects the bistable property of Opn5m, but it is too much to claim that the mutant mimics the property of Opn5L1.

Response: Thank you for your important comment. According to your suggestion, we weakened the claim about T188C mutant Opn5m throughout the manuscript. And we revised the sentences in the “Discussion” section as follows.

Page 14: “Therefore, it is suggested that the structural differences in the retinal binding pocket between Opn5L1 and T188C mutant Opn5m can alter the rate of the photocycle and the selectivity of the retinal isomerization. Additional mutations in Opn5m would be necessary to completely mimic the molecular properties of Opn5L1 by recapitulating the molecular evolutionary process from bistable opsin to photocyclic opsin.”

Comment 3: Page 10, lines 3-6: “The crystal structure of squid rhodopsin, which is a prototypical bistable opsin, shows that the residue at position 188 is accommodated in a β -sheet structure of the second extracellular loop and is located within 5 Å of the retinal polyene chain (Fig. S5a).”

How close are the sequences of squid rhodopsin and Opn5m? The authors should at least show the sequence alignment between squid rhodopsin and Opn5m, and they can also show the homology model of Opn5m based on the crystal structure of squid rhodopsin.

Response: According to your comment, we added the sequence comparison among bovine rhodopsin, squid rhodopsin, Opn5m and Opn5L1 in Fig. S1 and described the sequence similarity (28 %) between squid rhodopsin and *Xenopus* Opn5m in the legend of Fig. S1. We also showed the locations of three residues (positions 167, 188 and 212) in the crystal structure of squid rhodopsin (Fig. 1a).

Comment 4: The authors show the results of “Binding selectivity of retinal isomers” separately in Fig. 1 and 2, but this is very difficult to see. The authors can merge Fig. 1 and 2, keep action spectra to the main figure, and move the bar graphs to supplementary.

Response: According to your comment, we merged Figs. 1 and 2 as a new version of Fig. 2 and rearranged the supplementary figures.

Responses to Reviewer #2

Thank you very much for your critical reading of our manuscript and for your valuable comments. We revised the manuscript based on your comments and added page numbers and line numbers in the revised manuscript. According to the advice from a native English speaker, we tried to improve the English of our manuscript as much as possible. The following are our responses to your comments.

Comment 1: The reader needs to know what is meant by “all possible 19 mutations”. Coding/non-coding changes?

Response: The phrase “all possible 19 mutations” mentioned in our previous manuscript means that we replaced Thr188 with the other 19 natural amino acid residues. To clarify this point, we revised the sentence in the “Results” section as follows.

Page 8: “we replaced Thr188 with the other 18 natural amino acid residues in

addition to the cysteine of the T188C mutant protein.”

Comment 2: Change to read “wild type and mutant recombinant proteins”

Response: According to your comment, we revised the phrase throughout the manuscript.

Comment 3: What does this sentence mean? “Among these mutants, we detected the photopigments of seven mutants.....” Only these mutant proteins produced photopigments?

Response: We tried to obtain the recombinant Opn5m proteins whose residue at position 188 was replaced by the other 19 natural amino acid residues. We expressed these mutant recombinant proteins in cultured cells and reconstituted the photo-pigments by adding 11-cis or all-trans retinal to the collected cell membranes. Among these mutant proteins, we could produce the photo-pigments of T188C mutant protein (Fig. 1) and six other mutant proteins (T188S, T188G, T188A, T188N, T188M and T188Q) (Fig. 2), and could not detect the formation of the photo-pigments of the other 12 mutant proteins by the absorption spectrum analysis. To clarify this point, we revised the sentences in the “Results” section as follows.

Page 8: “To reveal the roles of the well-conserved threonine residue at position 188 in the Opn5m subgroup, we replaced Thr188 with the other 18 natural amino acid residues in addition to the cysteine of the T188C mutant protein. We expressed the mutant recombinant proteins in cultured cells and reconstituted the photo-pigments by adding 11-cis or all-trans retinal to the collected cell membranes. Among these mutant Opn5m, we could produce the

photo-pigments of six mutant proteins, T188S, T188G, T188A, T188N, T188M, and T188Q (Fig. 2), but could not detect the formation of the photo-pigments of the other 12 mutant proteins by absorption spectrum analysis.”

Comment 4: Say what form each retinal produces.

Response: In this experiment, we added 11-cis or all-trans retinal to the cell membranes which contained the mutant recombinant proteins. To clarify this point, we revised the sentences in the “Results” section as follows.

Page 5: “We reconstituted their photo-pigments by adding 11-cis or all-trans retinal to the collected cell membranes and purified them by affinity column chromatography using Rho1D4-conjugated agarose (Fig. 1b-1e).”

Page 8: “We expressed the mutant recombinant proteins in cultured cells and reconstituted the photo-pigments by adding 11-cis or all-trans retinal to the collected cell membranes.”

Comment 5: Change to read “wild-type and mutant Opn5m”. the grammatical structure used here and elsewhere in the text is very poor.

Response: According to your comment, we revised the phrase throughout the manuscript.

Comment 6: Define what is meant by “purified”.

Response: We prepared the recombinant Opn5m proteins which were tagged with the epitope sequence of the anti-bovine rhodopsin monoclonal antibody Rho1D4 at the C-terminus and purified them by affinity column chromatography

using Rho1D4-conjugated agarose. We already described the procedure in the “Materials and Methods” section. However, to clarify this point, we revised the sentences in the “Results” section as follows.

Page 5: “We reconstituted their photo-pigments by adding 11-cis or all-trans retinal to the collected cell membranes and purified them by affinity column chromatography using Rho1D4-conjugated agarose (Fig. 1b-1e).”

Comment 7: Need to state more clearly the illumination used in the experiments.

Response: According to your comment, we added the illumination wavelength in the legends of all the figures and described the illumination lamps and the optical filters used for the measurement of the absorption spectra in the “Materials and Methods” section as follows.

Page 16: “The sample was irradiated with light which was generated by a 1-kW tungsten halogen lamp (Master HILUX-HR, Rikagaku seiki) and passed through optical filters (Y-52 or UV-D36C, AGC Techno Glass). The transient absorption spectra of Opn5m T188C mutant protein were recorded using a Shimadzu UV2450 spectrophotometer at 10 ± 0.1 °C. The spectra were measured in the dark and after 2 min irradiation with light which was generated by a 1-kW tungsten halogen lamp and passed through an optical filter (Y-52, AGC Techno Glass). The transient absorption spectra of Opn5m T188C mutant protein were also recorded using a high-speed CCD camera spectrophotometer (C10000 system, Hamamatsu Photonics) kept at 37 ± 0.1 °C by a temperature controller (pqod, QUANTUM Northwest) to accelerate the photocyclic reaction, according to the methods used in our previous study for Opn5L1. The flash light for irradiation was generated by a short-arc power flash (SA-200, Nissin Electronic; pulse duration of ~ 170 μ s and flash lamp input of 200 J/F) and passed through

an optical filter (Y-48, AGC Techno Glass).”

Comment 8: Is it a “speculation” or a “conclusion”?

Response: In this study, we showed that wild-type Opn5m protein purified after the addition of 11-cis retinal exclusively contained 11-cis retinal. By contrast, wild-type Opn5m protein purified after the addition of all-trans retinal contained a substantial amount of 11-cis retinal in addition to all-trans retinal. Thus, we concluded that wild-type Opn5m protein preferentially binds 11-cis retinal. To clarify this point, we revised the sentence in the “Results” section as follows.

Page 6: “Thus, we concluded that wild-type Opn5m protein preferentially binds 11-cis retinal.”

Comment 9: Avoid using “mutants” – it is mutant protein of mutant Opn5m

Response: According to your comment, we revised the phrase throughout the manuscript.

Comment 10: The various experiments presented in the Figs need to be described in the Results –it is not sufficient to expect the reader to extract this from Fig legends and M&M.

Response: To clarify the relationship between the explanations in the main text and the figures, we inserted the figure numbers in several sentences of the main text and revised the sentences in the “Results” section as follows.

Page 8: “All the mutant proteins purified after the addition of all-trans retinal also had the absorbance in the visible region (black curve of Figs. 2g-2l). These

samples contained a smaller amount of 11-cis retinal (0~5 %) than wild-type (26 %, shown in Fig. 1b) in addition to a large amount of all-trans retinal (71~93 %) (right panels of Figs. 2g-2l). Yellow light irradiation of these samples induced the trans-cis isomerization of the retinal (right panels of Figs. 2g-2l) and shifted the absorption spectra into the UV region (red curve of Figs. 2g-2l). Moreover, all the mutant proteins purified after the addition of 11-cis retinal contained a larger amount of all-trans retinal (more than 17 %) than wild-type (5 %, shown in Fig. 1b) (right panels of Figs. 2a-2f). In particular, T188N and T188M mutant proteins contained predominantly all-trans retinal (85 % and 86 %, respectively) and a small amount of 11-cis retinal (3 % and 2 %, respectively) (Figs. 2d and 2e). The presence of the all-trans retinal bound form was confirmed by their yellow light-dependent spectral shift to the UV region (red curve of Figs. 2a-2f), which was not observed in wild-type (Fig. 1b)."

Comment 11: "Larger amount" – meaning what? Give a range if single figure not possible.

Response: According to your comment, we added the percentages of the retinal isomers throughout the manuscript.

Comment 12: "More preferentially" – bad grammar – either preferentially or not!

Response: According to your comment, we deleted the word "more" in the sentence of the "Results" section as follows.

Page 7: "T188C mutant protein of Opn5m preferentially binds all-trans retinal compared to wild-type"

Comment 13: “is responsible” – to say “is required”. Other sites may also be involve so to say that this is all down to one site may be going too far.

Response: According to your comment, we replaced the word “responsible” with “required” throughout the manuscript.

Comment 14: Use of “mutants” again.

Response: As mentioned in the response to comment 9, we revised the phrase throughout the manuscript.

Comment 15: Significance at temperature not explained.

Response: We set the temperature to 0 °C for the experiments shown in Figs. 1-3 and to 10 °C or 37 °C for the experiments shown in Fig. 4. To clarify this point, we revised the legends of the figures.

Comment 16: “all possible 19 mutations” – define

Response: As mentioned in the response to comment 1, we also revised the sentence in the “Discussion” section as follows.

Page 12: “we replaced Thr188 of Opn5m with the other 19 natural amino acid residues”

Comment 17: Move at least some of the description of the effect of the different mutations into the Results section.

Response: According to your comment, we added the following sentence in the “Results” section.

Page 8: “This showed that introduction of amino acid residues which have positively charged (Arg, His and Lys), negatively charged (Asp and Glu), bulky aromatic (Phe, Trp and Tyr), cyclic (Pro) or branched (Ile, Leu and Val) side chains into this position prevented the formation of the photo-pigments of Opn5m.”

Comment 18: Comment on Figure 1 and 2: Fig 1 is very difficult to follow. I presume the 1 and 2 labelling of spectra refer to 11-cis and all trans. Suggest that the mutation is put on Figure alongside a – d. Bar charts may be easier to understand, but need much more description in Results section. Figs 1 and 2 need to be combined into single figure.

Response: According to your comment, we merged Figs. 1 and 2 as a new version of Fig. 2 and rearranged the supplementary figures. We also changed the design of bar charts to clearly show the ratios of the retinal isomers in the figures. And we added the percentages of the retinal isomers in the main text.

Comment 19: Comment on Figure 3: put wild-type and mutations on Figure, not just in legend.

Response: According to your comment, we revised the figures to clearly show the labels of wild-type and mutations.

REVIEWERS' COMMENTS:

Reviewer #1 (Remarks to the Author):

I thank the authors for their revisions to this work. The authors have addressed all my concerns by carrying out additional experiments, and I believe the work is now suitable for publication.